# Scalable Methods for 8-bit Training of Neural Networks

**Ron Banner**[1]*, **Itay Hubara**[2]*, **Elad Hoffer**[2]*, **Daniel Soudry**[2]
{itayhubara, elad.hoffer, daniel.soudry}@gmail.com
{ron.banner}@intel.com

(1) Intel - Artificial Intelligence Products Group (AIPG)
(2) Technion - Israel Institute of Technology, Haifa, Israel

## Abstract

Quantized Neural Networks (QNNs) are often used to improve network efficiency during the inference phase, i.e. after the network has been trained. Extensive research in the field suggests many different quantization schemes. Still, the number of bits required, as well as the best quantization scheme, are yet unknown. Our theoretical analysis suggests that most of the training process is robust to substantial precision reduction, and points to only a few specific operations that require higher precision. Armed with this knowledge, we quantize the model parameters, activations and layer gradients to 8-bit, leaving at a higher precision only the final step in the computation of the weight gradients. Additionally, as QNNs require batch-normalization to be trained at high precision, we introduce Range Batch-Normalization (BN) which has significantly higher tolerance to quantization noise and improved computational complexity. Our simulations show that Range BN is equivalent to the traditional batch norm if a precise scale adjustment, which can be approximated analytically, is applied. To the best of the authors' knowledge, this work is the first to quantize the weights, activations, as well as a substantial volume of the gradients stream, in all layers (including batch normalization) to 8-bit while showing state-of-the-art results over the ImageNet-1K dataset.

## 1 Introduction

Deep Neural Networks (DNNs) achieved remarkable results in many fields making them the most common off-the-shelf approach for a wide variety of machine learning applications. However, as networks get deeper, using neural network (NN) algorithms and training them on conventional general-purpose digital hardware is highly inefficient. The main computational effort is due to massive amounts of multiply-accumulate operations (MACs) required to compute the weighted sums of the neurons' inputs and the parameters' gradients.

Much work has been done to reduce the size of networks. The conventional approach is to compress a trained (full precision) network [4, 19, 12] using weights sharing, low rank approximation, quantization, pruning or some combination thereof. For example, Han et al., 2015 [7] successfully pruned several state-of-the-art large-scale networks and showed that the number of parameters can be reduced by an order of magnitude.

Since training neural networks requires approximately three times more computation power than just evaluating them, quantizing the gradients is a critical step towards faster training machines. Previous

---

work demonstrated that by quantizing network parameters and intermediate activations during the training phase more computationally efficient DNNs could be constructed.

Researchers [6, 5] have shown that 16-bit is sufficient precision for most network training but further quantization (i.e., 8-bit) results with severe degradation. Our work is the first to almost exclusively train at 8-bit without harming classification accuracy. This is addressed by overcoming two main obstacles known to hamper numerical stability: batch normalization and gradient computations.

The traditional batch normalization [11] implementation requires the computation of the sum of squares, square-root and reciprocal operations; these require high precision (to avoid zero variance) and a large dynamic range. It should come as no surprise that previous attempts to use low precision networks did not use batch normalization layers [21] or kept them in full precision [24]. This work replaces the batch norm operation with range batch-norm (range BN) that normalizes inputs by the range of the input distribution (i.e., $\max(x) - \min(x)$). This measure is more suitable for low-precision implementations. Range BN is shown analytically to approximate the original batch normalization by multiplying this range with a scale adjustment that depends on the size of the batch and equals to $(2 \cdot \ln(n))^{-0.5}$. Experiments on ImageNet with Res18 and Res50 showed no distinguishable difference between accuracy of Range BN and traditional BN.

The second obstacle is related to the gradients quantization. Given an upstream gradient $g_l$ from layer $l$, layer $l-1$ needs to apply two different matrix multiplications: one for the layer gradient $g_{l-1}$ and the other for the weight gradient $g_W$ which are needed for the update rule. Our analysis indicates that the statistics of the gradient $g_l$ violates the assumptions at the crux of common quantization schemes. As such, quantizing these gradients constitutes the main cause of degradation in performance through training. Accordingly, we suggest to use two versions of layer gradients $g_l$, one with low-precision (8-bit) and another with higher-precision (16-bit). The idea is to keep all calculations with $g_l$ that does not involve a performance bottleneck at 16 bits, while the rest at 8 bits. As the gradients $g_W$ are required only for the weight update, they are computed using the 16 bits copy of $g_l$. On the other hand, the gradient $g_{l-1}$ is required for the entire backwards stream and as such it is computed using the corresponding 8-bit version of $g_l$. In most layers of the DNN these computations can be performed in parallel. Hence $g_W$ can be computed at high precision in parallel with $g_{l-1}$, without interrupting the propagation of $g_l$ to lower layers. We denote the use of two different arithmetic precision operations in the differentiation process as "Gradients Bifurcation".

## 2    Previous Work

While several works [6, 5] have shown that training at 16-bit is sufficient for most networks, more aggressive quantization schemes were also suggested [24, 16, 14, 10]. In the extreme case, the quantization process used only one bit which resulted in binarized neural networks (BNNs) [9] where both weights and activations were constrained to -1 and 1. However, for more complex models and challenging datasets, the extreme compression rate resulted in a loss of accuracy. Recently, Mishra et al. [15] showed that this accuracy loss can be prevented by merely increasing the number of filter maps in each layer, thus suggesting that quantized neural networks (QNNs) do not possess an inherent convergence problem. Nevertheless, increasing the number of filter maps enlarge quadratically the number of parameters, which raises questions about the efficiency of this approach.

In addition to the quantization of the forward pass, a growing interest is directed towards the quantization of the gradient propagation in neural networks. A fully quantized method, allowing both forward and backward low-precision operations will enable the use of dedicated hardware, with considerable computational, memory, and power benefits. Previous attempts to discretize the gradients managed to either reduce them to 16-bit without loss of accuracy [5] or apply a more aggressive approach and reduce the precision to 6-8 bit [24, 10] with a noticeable degradation. Batch normalization is mentioned by [21] as a bottleneck for network quantization and is either replaced by a constant scaling layer kept in full precision, or avoided altogether; this clearly has some impact on performance (e.g., AlexNet trained over ImageNet resulted with top-1 error of $51.6\%$, where the state of the art is near $42\%$) and better ways to quantize normalization are explicitly called for. Recently L1 batch norm with only linear operations in both forward and backward propagation was suggested by [22, 8] with improved numerical stability. Yet, our experiments show that with 8-bit training even L1 batch norm is prone to overflows when summing over many large positive values. Finally, Wen

et al. [20] focused on quantizing the gradient updates to ternary values to reduce the communication bandwidth in distributed systems.

We claim that although more aggressive quantization methods exist, 8-bit precision may prove to have a "sweet-spot" quality to it, by enabling training with no loss of accuracy and without modifying the original architecture. Moreover, we note that 8-bit quantization is better suited for future and even current hardware, many of which can already benefit from 8-bit operations [17]. So far, to the best of our knowledge, no work has succeeded to quantize the activations, weights, and gradient of all layers (including batch normalization) to 8-bit without any degradation.

## 3   Range Batch-Normalization

For a layer with $n \times d-$dimensional input $x = (x^{(1)}, x^{(2)}, ..., x^{(d)})$, traditional batch norm normalizes each dimension

$$\hat{x}^{(d)} = \frac{x^{(d)} - \mu^d}{\sqrt{\text{Var}[x^{(d)}]}}, \tag{1}$$

where $\mu^d$ is the expectation over $x^{(d)}$, $n$ is the batch size and $\text{Var}[x^{(d)}] = \frac{1}{n}||x^{(d)} - \mu^d||_2^2$. The term $\sqrt{\text{Var}[x^{(d)}]}$ involves sums of squares that can lead to numerical instability as well as to arithmetic overflows when dealing with large values. The Range BN method replaces the above term by normalizing according to the range of the input distribution (i.e., $\max(\cdot) - \min(\cdot)$), making it more tolerant to quantization. For a layer with $d-$dimensional input $x = (x^{(1)}, x^{(2)}, ..., x^{(d)})$, Range BN normalizes each dimension

$$\hat{x}^{(d)} = \frac{x^{(d)} - \mu^d}{C(n) \cdot \text{range}(x^{(d)} - \mu^d)}, \tag{2}$$

where $\mu^d$ is the expectation over $x^{(d)}$, $n$ is the batch size, $C(n) = \frac{1}{\sqrt{2 \cdot \ln(n)}}$ is a scale adjustment term, and $\text{range}(x) = \max(x) - \min(x)$.

The main idea behind Range BN is to use the scale adjustment $C(n)$ to approximate the standard deviation $\sigma$ (traditionally being used in vanilla batch norm) by multiplying it with the range of the input values. Assuming the input follows a Gaussian distribution, the range (spread) of the input is highly correlated with the standard deviation magnitude. Therefore by normalizing the range by $C(n)$ we can estimate $\sigma$. Note that the Gaussian assumption is a common approximation (e.g., Soudry et al. [18]), based on the fact that the neural input $x^{(d)}$ is a sum of many inputs, so we expect it to be approximately Gaussian from the central limit theorem.

We now turn to derive the normalization term $C(n)$. The expectation of maximum of Gaussian random variables are bounded as follows [13]:

$$0.23\sigma \cdot \sqrt{\ln(n)} \leq E[\max(x^{(d)} - \mu^d)] \leq \sqrt{2}\sigma\sqrt{\ln(n)}. \tag{3}$$

Since $x^{(d)} - \mu^d$ is symmetrical with respect to zero (centred at zero and assumed gaussian), it holds that $E[\max(\cdot)] = -E[\min(\cdot)]$; hence,

$$0.23\sigma \cdot \sqrt{\ln(n)} \leq -E[\min(x^{(d)} - \mu^d)] \leq \sqrt{2}\sigma\sqrt{\ln(n)}. \tag{4}$$

Therefore, by summing Equations 3 and 4 and multiplying the three parts of the inequality by the normalization term $C(n)$, Range BN in Eq. 2 approximates the original standard deviation measure $\sigma$ as follows:

$$0.325\sigma \leq C(n) \cdot \text{range}(x^{(d)} - \mu^d) \leq 2 \cdot \sigma$$

Importantly, the scale adjustment term $C(n)$ plays a major role in RangeBN success. The performance was degraded in simulations when $C(n)$ was not used or modified to nearby values.

## 4   Quantized Back-Propagation

**Quantization methods:**   Following [23] we used the GEMMLOWP quantization scheme as described in Google's open source library [1]. A detailed explanation of this approach is given in Appendix.While GEMMLOWP is widely used for deployment, to the best of the authors knowledge this is the first time GEMMLOWP quantization is applied for training. Note that the activations maximum and minimum values were computed by the range BN operator, thus finding the normalization scale (see Appendix)does not require additional $O(n)$ operations.

Finally we note that a good convergence was achieved only by using stochastic rounding [6] for the gradient quantization. This behaviour is not surprising as the gradients will serve eventually for the weight update thus unbiased quantization scheme is required to avoid noise accumulation.

**Gradients Bifurcation:**   In the back-propagation algorithm we recursively calculate the gradients of the loss function $\mathcal{L}$ with respect to $I_\ell$, the input of the $\ell$ neural layer,

$$g_\ell = \frac{\partial \mathcal{L}}{\partial I_\ell}, \tag{5}$$

starting from the last layer. Each layer needs to derive two sets of gradients to perform the recursive update. The layer activation gradients:

$$g_{\ell-1} = g_\ell W_\ell^T, \tag{6}$$

served for the Back-Propagation (BP) phase thus passed to the next layer,and the weights gradients

$$g_{W_\ell} = g_\ell I_{\ell-1}^T, \tag{7}$$

used to updated the weights in layer $\ell$. Since the backward pass requires twice the amount of multiplications compared to the forward pass, quantizing the gradients is a crucial step towards faster training machines. Since $g_\ell$, the gradients streaming from layer $\ell$, are required to compute $g_{\ell-1}$, it is important to expedite the matrix multiplication described in Eq.6. The second set of gradient derive in Eq.7 is not required for this sequential process and thus we choose to keep this matrix multiplication in full precision. We argue that the extra time required for this matrix multiplication is comparably small to the time required to communicate the gradients $g_\ell$. Thus, in this work the gradients used for the weight gradients derivation are still in float. In section 6, we show empirically that bifurcation of the gradients is crucial for high accuracy results.

**Straight-Through Estimator:**   Similar to previous work [9, 15], we used the straight-through estimator (STE) approach to approximate differentiation through discrete variables. This is the most simple and hardware friendly approach to deal with the fact that the exact derivative of discrete variables is zero almost everywhere.

## 5   When is quantization of neural networks possible?

This section provides some of the foundations needed for understanding the internal representation of quantized neural networks. It is well known that when batch norm is applied after a convolution layer, the output is invariant to the norm of the weight on the proceeding layer [11] i.e., $BN(C \cdot W \cdot x) = BN(W \cdot x)$ for any given constant $C$. This quantity is often described geometrically as the norm of the weight tensor, and in the presence of this invariance, the only measure that needs to be preserved upon quantization is the *directionality* of the weight tensor. In the following we show that quantization preserves the direction (angle) of high-dimensional vectors when $W$ follows a Gaussian distribution.

More specifically, for networks with $M$-bit fixed point representation, the angle is preserved when the number of quantization levels $2^M$ is much larger than $\sqrt{2\ln(N)}$, where $N$ is the size of quantized vector. This shows that significant quantization is possible on practical settings. Taking for example the dimensionality of the joint product in a batch with 1024 examples corresponding to the last layer of ResNet-50, we need no more than 8-bit of precision to preserve the angle well (i.e., $\sqrt{2\ln(3 \cdot 3 \cdot 2048 \cdot 1024)} = 5.7 << 2^8$). We stress that this result heavily relays on values being distributed according to a Gaussian distribution, and suggests why some vectors are robust to quantization (e.g., weights and activations) while others are more fragile (e.g., gradients).

## 5.1 Problem Statement

Given a vector of weights $W = (w_0, w_1, ..., w_{N-1})$, where the weights follow a Gaussian distribution $W \sim N(0, \sigma)$, we would like to measure the cosine similarity (i.e., cosine of the angle) between $W$ and $Q(W)$, where $Q(\cdot)$ is a quantization function. More formally, we are interested in estimating the following geometric measure:

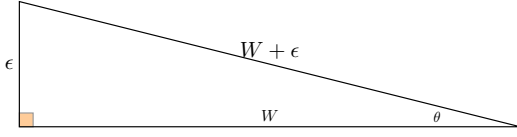

Figure 1: Graphic illustration of the angle between full precision vector $W$ and its low precision counterpart which we model as $W + \epsilon$ where $\epsilon \sim \mathbb{U}(-\Delta/2, \Delta/2)$

$$\cos(\theta) = \frac{W \cdot Q(W)}{||W||_2 \cdot ||Q(W)||_2} \tag{8}$$

We next define the quantization function $Q(\cdot)$ using a fixed quantization step between adjacent quantified levels as follows:

$$Q(x) = \Delta \cdot \left( \left\lfloor \frac{x}{\Delta} \right\rfloor + \frac{1}{2} \right), \text{ where } \Delta = \frac{\max(|W|)}{2^M} \tag{9}$$

We consider the case where quantization step $\Delta$ is much smaller than mean($|W|$). Under this assumption correlation between $W$ and quantization noise $W - Q(W) = (\epsilon_0, \epsilon_1, ..., \epsilon_{N-1})$ is negligible, and can be approximated as an additive noise. Our model assumes an additive quantization noise $\bar{\epsilon}$ with a uniform distribution i.e., $\epsilon_i \sim \mathcal{U}[-\Delta/2, \Delta/2]$ for each index $i$. Our goal is to estimate the angle between $W$ and $W + \bar{\epsilon}$ for high dimensions (i.e., $N \to \infty$).

## 5.2 Angle preservation during quantization

In order to estimate the angle between $W$ and $W + \epsilon$, we first estimate the angle between $W$ and $\epsilon$. It is well known that if $\epsilon$ and $W$ are independent, then at high dimension the angle between $W$ and $\epsilon$ tends to $\frac{\pi}{2}$ [2] i.e., we get a right angle triangle with $W$ and $\epsilon$ as the legs, while $W + \epsilon$ is the hypotenuse as illustrated in Figure 1-right. The cosine of the angle $\theta$ in that triangle can be approximated as follows:

$$\cos(\theta) = \frac{||W||}{||W + \epsilon||} \geq \frac{||W||}{||W|| + ||\epsilon||} \tag{10}$$

Since $W$ is Gaussian, we have that $E(||W||) \cong \sqrt{N}\sigma$ in high dimensions [3]. Additionally, in Appendix **??** we show that $E(||\bar{\epsilon}||) \leq \sqrt{N/12} \cdot \Delta$. Moreover, at high dimensions, the relative error made as considering $E||X||$ instead of the random variable $||X||$ becomes asymptotically negligible [2]. Therefore, the following holds in high dimensions:

$$\cos(\theta) \geq \frac{\sigma}{\sigma + E(\Delta)/\sqrt{12}} = \frac{2^M \cdot \sigma}{2^M \cdot \sigma + E(\max(|W|))/\sqrt{12}} \tag{11}$$

Finally, $E(\max(W)) \leq \sqrt{2}\sigma\sqrt{\ln(N)}$ when $W$ follows a Gaussian distribution [13], establishing the following:

$$\cos(\theta) \geq \frac{2^M}{2^M + \sqrt{\ln N}/\sqrt{6}} \tag{12}$$

Eq. 12 establishes that when $2^M >> \sqrt{\ln(N)}$ the angle is preserved during quantization. It is easy to see that in most practical settings this condition holds even for challenging quantizations. Moreover, this results highly depends on the assumption made about the Gaussian distribution of $W$ (transition from equation 11 to equation 12).

## 6 Experiments

We evaluated the ideas of Range Batch-Norm and Quantized Back-Propagation on multiple different models and datasets. The code to replicate all of our experiments is available on-line [2].

## 6.1 Experiment results on cifar-10 dataset

To validate our assumption that the cosine similarity is a good measure for the quality of the quantization, we ran a set of experiments on Cifar-10 dataset, each with a different number of bits, and then plotted the average angle and the final accuracy. As can be seen in Figure 2 there is a high correlation between the two. Taking a closer look the following additional observations can be made: (1) During quantization the direction of vectors is better preserved with the forward pass compared to the backward pass; (2) validation accuracy follows tightly the cosine of the angle in the backward pass, indicating gradient quantization as the primary bottleneck; (3) as expected, the bound on $E(\cos(\theta))$ in Eq. 12 holds in the forward pass, but less so in the backward pass, where the Gaussian assumption tends to break. The histograms in Figure 2 further confirms that the layer gradients $g_l$ do not follow Gaussian distribution. These are the values that are bifurcated into low and high precision copies to reduce noise accumulation.

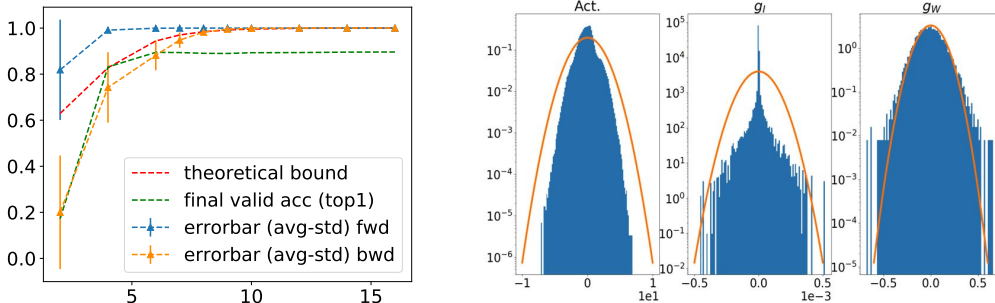

Figure 2: Left: empirical and theoretical analysis of cosine similarity, $\cos(\theta)$, with respect the number of bits used for quantization. Right: Histograms of activations, layer gradients $g_l$ and weight gradients $g_W$. To emphasize that $g_l$ do not follow a Gaussian distribution, the histograms were plotted in a log-scale (ResNet-18, Cifar-10).

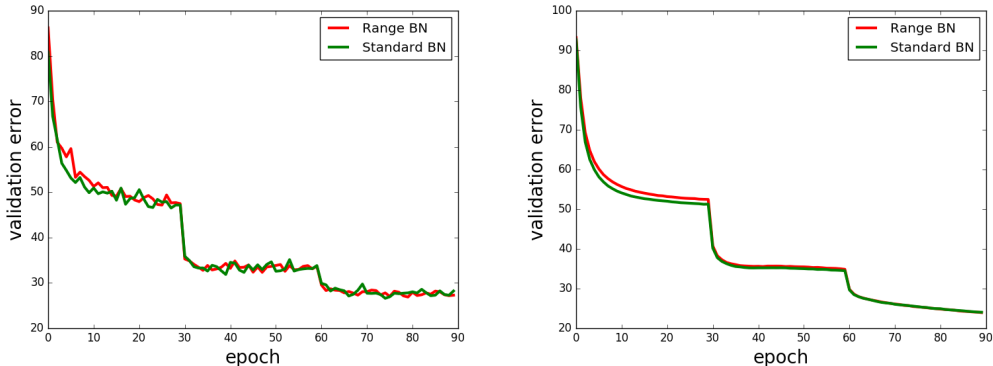

Figure 3: Equivalent accuracy with standard and range batch-norm (ResNet-50, ImageNet).

## 6.2 Experiment results on ImageNet dataset: Range Batch-Normalization

We ran experiments with Res50 on ImageNet dataset showing the equivalence between the standard batch-norm and Range BN in terms of accuracy. The only difference between the experiments was the use of Range BN instead of the traditional batch-norm. Figure 3 compares between the two. It shows equivalence when models are trained at high precision. We also ran simulations on other datasets and models. When examining the final results, both were equivalent i.e., 32.5% vs 32.4% for ResNet-18 on ImageNet and 10.5% vs 10.7% for ResNet-56 on Cifar10. To conclude, these simulations prove that we can replace standard batch-norm with Range BN while keeping accuracy unchanged. Replacing the sum of squares and square root operations in standard batch-norm by a few maximum and minimum operations has a major benefit in low-precision implementations.

## 6.3 Experiment results on ImageNet dataset: Putting it all together

We conducted experiments using RangeBN together with Quantized Back-Propagation. To validate this low precision scheme, we were quantizing the vast majority of operations to 8-bit. The only operations left at higher precising were the updates (float32) needed to accumulate small changes from stochastic gradient descent, and a copy of the layer gradients at 16 bits needed to compute $g_W$. Note that the float32 updates are done once per minibatch while the propagations are done for each example (e.g., for a minibatch of 256 examples the updates constitute less than 0.4% of the training effort). Figure 4 presents the result of this experiment on ImageNet dataset using ResNet18 and ResNet50. We provide additional results using more aggressive quantizations in Appedix F.

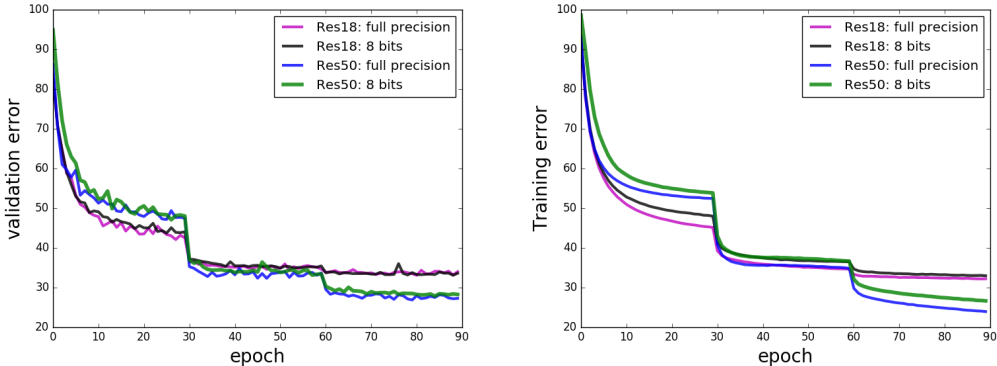

Figure 4: Comparing a full precision run against 8-bit run with Quantized Back-Propagation and Range BN (ResNet-18 and ResNet-50 trained on ImageNet).

## 7 Discussion

In this study, we investigate the internal representation of low precision neural networks and present guidelines for their quantization. Considering the preservation of direction during quantization, we analytically show that significant quantization is possible for vectors with a Gaussian distribution. On the forward pass the inputs to each layer are known to be distributed according to a Gaussian distribution, but on the backward pass we observe that the layer gradients $g_l$ do not follow this distribution. Our experiments further assess that angle is not well preserved on the backward pass, and moreover final validation accuracy tightly follows that angle. Accordingly, we bifurcate the layer gradients $g_l$ and use it at 16-bits for the computation of the weight gradient $g_W$ while keeping the computation of next layer gradient $g_{l-1}$ at 8-bit. This enables the (slower) 16-bits computation of $g_W$ to be be done in parallel with $g_{l-1}$, without interrupting the propagation the layer gradients.

We further show that Range BN is comparable to the traditional batch norm in terms of accuracy and convergence rate. This makes it a viable alternative for low precision training. During the forward-propagation phase computation of the square and square root operations are avoided and replaced by $\max(\cdot)$ and $\min(\cdot)$ operations. During the back-propagation phase, the derivative of $\max(\cdot)$ or $\min(\cdot)$ is set to one where the coordinates for which the maximal or minimal values are attained, and is set to zero otherwise.

Finally, we combine the two novelties into a single training scheme and demonstrate, for the first time, that 8-bit training on a large scale dataset does not harm accuracy. Our quantization approach has major performance benefits in terms of speed, memory, and energy. By replacing float32 with int8, multiplications become 16 times faster and at least 15 times more energy efficient [10]. This impact is attained for 2/3 of all the multiplications, namely the forward pass and the calculations of the layer gradients $g_l$. The weight gradients $g_W$ are computed as a product of 8-bit precision (layer input) with a 16-bit precision (unquantized version of $g_l$), resulting with a speedup of x8 for the rest of multiplications and at least x2 power savings. Although previous works considered an even lower precision quantization (up-to 1-bit), we claim that 8-bit quantization may prove to be more of an interest. Furthermore, 8-bit matrix multiplication is available as an off-the-shelf operation in existing hardware and can be easily adopted and used with our methods.

## Acknowledgments

This research was supported by the Israel Science Foundation (grant No. 31/1031), and by the Taub foundation. A Titan Xp used for this research was donated by the NVIDIA Corporation. The authors are pleased to acknowledge that the work reported in this paper was substantially performed at Intel - Artificial Intelligence Products Group (AIPG).

## Footnotes

[2]https://github.com/eladhoffer/quantized.pytorch

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
