[Supplementary Material · Supplementary.pdf]

# Supplementary material

## A Bound on the expected norm of $\epsilon$

By Jensen inequality the following holds true:

$$E(||\epsilon||) = E\left[\sqrt{\sum_{i=0}^{N-1} \epsilon_i^2}\right] \leq \sqrt{E\left[\sum_{i=0}^{N-1} \epsilon_i^2\right]} = \sqrt{\sum_{i=0}^{N-1} E[\epsilon_i^2]} \tag{1}$$

Given a uniform random variable in $\epsilon_i \sim U[-\Delta/2, \Delta/2]$, we next derive the expected value of its square as follows:

$$E[\epsilon_i^2] = \int_{-\Delta/2}^{\Delta/2} x^2 \cdot \frac{1}{\Delta} \cdot dx = \frac{\Delta^2}{12} \tag{2}$$

Note that $\Delta$ is a random variable. Hence, we substitute Eq 2 into Eq 1 using the following conditional expectation:

$$E(||\epsilon|| \text{ given } \Delta) \leq \sqrt{\frac{N}{12}} \cdot \Delta \tag{3}$$

We can now establish $E(||\epsilon||)$ as follows:

$$E(||\epsilon||) \leq E(E( ||\epsilon|| |\Delta))|\Delta)) = \sqrt{\frac{N}{12}} \cdot E(\Delta) \tag{4}$$

## B Quantization methods

Following [10] we choose to use the GMMLOWP quantization scheme as decribed in Google's open source library [2]. Given an input tensor $x$, clamping values $[v_{\min}, v_{\max}]$,and number of bits $M$ we set the output to be:

$$\text{scale} = (v_{\max} - v_{\min})/2^M$$
$$\text{zero} - \text{point} = \text{round}(\min(\max(-v_{\min}/\text{scale}, 0), 2^M))$$
$$\text{output} = \text{round}(x/\text{scale} + \text{zero} - \text{point})$$

The clamping values for the weights and activations were defined as the input's absolute maximum and minimum. Since the activations can have a high dynamic range which can be aggressively clamped as shown by [10] we defined its clamping values to be the average of absolute maximum and minimum values of K chunks. This reduces the dynamic range Variance and allows smaller quantization steps.

it is important to note that a good convergence was achieved only by using stochastic rounding [4]. This behaviour is not surprisings as the gradients serves eventually for the weight update thus unbias quantization scheme is required to avoid quantization noise accumulation.

## C Additional Experiments

In this section we present our more aggressive quantization experiments of the Quantized Back-Propagation scheme. In the extreme case, QBP ternarizes the gradients and uses only 1-bit for the weights and activations. In this case, we refer to QBP networks as Ternarized Back-Propagation (TBP), in which all forward MACs operations can be replaced with XNOR and population count (*i.e.,* counting the number of ones in the binary number) operations. To avoid significant degradation in test accuracy, we apply stochastic ternarization and increase the number of filter maps in a each convolution layer.

## C.1 CIFAR10

A well studied dataset is the CIFAR10 image classification benchmark first introduced by Krizhevsky [7]. CIFAR10 consists of a training set of size 50K, and a test set of size 10K color images. Here, each images represents one of the following categories: airplanes, automobiles, birds, cats, deer, dogs, frogs, horses, ships and trucks.

We trained a VGG-like network similar to the one suggested by Hubara et al. [6] on the CIFAR10 dataset, with the same hyper-parameters used in the original work. We compared two variants: the original model BNN model and the BNN model trained with TBP (this paper), where ternarized gradients are used.

The results shown in Table (2) demonstrate that if we inflate the convolutional filters by 3 we can achieve similar results as the BNN and full precision models achieved. This is in accordance with previous finding [9], that found that widening the network can mitigate accuracy drop inflicted by low precision training. To make sure this is not a unique case for BNN we also applied TBP on ResNet with depth of 18. As can be seen from Table (2), as before, inflating the network improves performance, until it is only 1% from the original performance, after inflating it by 5.

## C.2 ImageNet

Next, we applied TBP to the more challenging ImageNet classification task introduced by Deng et al. [3]. It consists of a training set of size 1.2M samples and a test set of size 50K. Each instance is labeled with one of 1000 categories including objects, animals, scenes, and even some abstract shapes. We report two error rates for this dataset: top-1 and top-5, as is typical done. Top-$k$ error rate represent the fraction of test images for which the correct label is not among the $k$ most probable labels predicted by the model.

We run several experiments on AlexNet inflated by 3. Similarly to previous work and to ease the comparison we kept first and last layer in full precision. With binarized weights, 4-bit activations and gradients, TBP converged to 53.3% top-1 accuracy and 75.84% top-5 accuracy. By using only 2bit activation TBP reached 49.6% top-1 accuracy and 73.1% top-5 accuracy. We are currently working on more advanced typologies such as ResNet-50 model [5]. Results are summarized in Table (1).

Table 1: Classification top-1 validation error rates of TBP BNNs trained on ImageNet with AlexNet topology. I, A, W, G stands for Inflation, Activation bits, Weights bits and Gradient bits respectively

| MODEL | I | A | W | G | ERROR |
|-------|---|---|---|---|-------|
| TBP (OURS) | 3 | 2 | 1 | 4 | 50.43% |
| TBP (OURS) | 3 | 4 | 1 | 4 | 46.7% |
| DOREFA (ZHOU ET AL.) | 1 | 2 | 1 | 6 | 53.9% |
| DOREFA (ZHOU ET AL.) | 1 | 4 | 1 | 6 | 51.8% |
| WRPN (MISHRA ET AL.) | 2 | 1 | 1 | 32 | 48.3% |
| WRPN (MISHRA ET AL.) | 2 | 2 | 2 | 32 | 55.8% |
| SINGLE PRECISION -NO QUANTIZTION | | | | | |
| ALEXNET (KRIZHEVSKY) | 1 | 32 | 32 | 32 | 43.5% |

## C.3 Additional experiments

To shed light on why TBP works, and on what is yet to be solved, we conducted additional set of experiments on the CIFAR-10 dataset with both ResNet-18 and the BNN like typologies. The Back Propagation algorithm includes three phases. The forward phase in which we calculate the activations values for each layer. The Backward-Propagation (BP) phase in which the gradients from layer $\ell$ pass to layer $\ell - 1$ and the update phase in which the model parameters (*i.e.,* weights) are updated. Both stages require MAC operations as detailed in section **??**. In this paper we focused on the BP stage. As oppose to the update stage that can be done in parallel, BP is a sequential stage. However, if the update phase uses full precision MAC operation the hardware need to support it. Moreover compressing the update gradients reduce the communication bandwidth in distributed systems. Thus, quantizing the weights gradients for the updates phase can also reduce power consummation and accelerate the training.

Table 2: Classification test error rates of TBP BNNs trained on CIFAR10

| MODEL | ERROR RATE |
|---|---|
| BINARIZED ACTIVATIONS,WEIGHTS AND TERN GRADIENTS | |
| TBP BNN, INFLATED BY 3 | 9.53% |
| TBP RESNET-18, INFLATED BY 5 | 10.8% |
| TBP RESNET-18, INFLATED BY 3 | 14.21% |
| TBP RESNET-18 [5] | 18.5% |
| BINARIZED ACTIVATIONS,WEIGHTS | |
| BNN [6] | 10.15% |
| BINARIZED RESNET-18, INFLATED BY 5 | 10.7% |
| NO BINARIZATION (STANDARD RESULTS) | |
| BNN [6] | 10.94% |
| RESNET-18 | 9.64% |

**Ternarizing both stages.** Ternarizing both stages results with completely MAC free training. However, our results show that without enabling at least 3bit precision for the update stage the model reaches only approximately 80% accuracy. This indicates that the ternarization noise is too high, and thus distorts the update gradients direction. If we stop the gradients ternarization once the accuracy ceases to increase, the convergence continues and the accuracy increases to the same accuracy as TBP. Thus, ternarizing the update stage can be used to accelerate TBP training of BNN networks by first training it with ternarized weights gradients and then, for the last couple of epochs, continue training with full precision weights gradients.

---

**Algorithm 1:** Multiple stochastic ternarization sampling algorithm.

---

**Require:** gradients from previous layer $g_{s_k}$, binarized activation $a_{k-1}^b$ and number of samples $S$.
  **for** $k = 1$ to $S$ **do**
    $g_{s_k}^b \leftarrow \text{StcTern}(g_{s_k})$
    $g_{W_k^b} \mathrel{+}= g_{s_k}^b a_{k-1}^b$
  **end for**
  **Return** $g_{W_k^b} \mathrel{/}= S$

---

**Multiple stochastic ternarization sampling.** To alleviate the need for float MAC operation in the update phase we suggest to use gated XNOR operation multiple times, each time with different stochastic sample of the ternarized tensor and average the results. The algorithm is detailed in Algorithm (1) and results are given in Table (3). As expected the accuracy improves with the amount of sampling. To find the number of samples needed for each layer we adopted a similar geometrical approach as suggested by Anderson & Berg [1] and measured the correlation coefficient ($R$) between

| Data set | Error |
|---|---|
| TBP BNN, 1 sample | 20.1% |
| TBP BNN, 5 samples | 13.5% |
| TBP BNN, 10 samples | 13% |
| TBP BNN, 20 samples | 12% |
| TBP BNN, $R > 0.7$ | 12.5% |
| ResNet-18 10 samples | 14% |
| ResNet-18 20 samples | 12.7% |

Table 3: Classification test error rates of TBP BNNs trained on CIFAR10 with Multiple Stochastic ternarization sampling for the update phase. ResNet-18 and BNN models were inflated by 5 and 3 respectively.

the update gradients received with and without ternarization. Our experiments indicate that more samples are required for the first two convolution layer (12 samples) while the rest of the layers need approximately 6 samples. Using this configuration keeps the correlation coefficient above 0.7 and results with 87.5% accuracy.