[Reviews · NeurIPS 2018]

Reviewer 1



# Summary of the paper The goal of this paper is to train and quantize a model into 8 bit. This is interesting given the fact that most of the existing works are based on 16 bit and people are having some difficulties in training 8bit models. The paper identified that the training difficulty comes from batchnorm and it proposed a variant of batchnorm called range batchnorm which alleviate the numerical instability of the original batchnorm occurring with the quantized models. By such simple modification, the paper shows that a 8 bit model can be easily trained using GEMMLOWP, an existing framework. The paper also tried to analyze and understand the proposed approach in a theoretical manner. Experiments well supported the argument of the paper. # General comments I am on the positive because I found the paper has a clean goal (training 8 bit model), identified the right problem (batch norm) and proposed a solution to address the problem (range batch norm). The paper is technically sound and I appreciate the authors’ effort in understanding of the problem which puts some foundation of the proposed solution. # Quality The proposed method is technically sound, simple and effective. I roughly checked all the equations which look good in general. 1- Given this is a model quantization paper, I would be interested in a evaluation and comparison on the model size and speed. 2- The analysis in section 3 is good. However, the assumption that x^{(d)} is gaussian distributed is probably not true in the real scenario. The input data could be Gaussian however, the input to other following layers could often not be. But I don’t think this is a severe problem for this paper given the fact that properly analyzing neural networks is still a challenging theoretical problem. 3- Section 5 derives the lower bound of the expectation of cosine distance. But how about the variance of the cosine? I think the variance could also be an important metric to understand better about such performance guarantee. # Clarity The paper is well written and easy to follow. Few comments: 1- Appendix E is an important technical detail and should be included in the main body (section 4) of the paper. If you feel the paper is too long, I would suggest reducing Section 5 a little bit, e.g., Figure 1-right does not seem to add additional information while it took a lot of space. 2- Fix typos, e.g., Figure 1-left the x label “treshold” -> “threshold”; Line 233 “Res50” -> “ResNet-50”. Please be consistent with the terminologies and short forms. The caption of figure 2, “with respect the” -> “with respect to the”. 3- All equations should be properly punctuated. # Originality I believe the Range Batchnorm and a systematic method to quantize models into 8 bit are novel. # Significance I think the results presented in this paper could be interesting to researchers in theory and quantization. Quantizing a model into 8 bit is interesting which might inspire many more interesting future work in this area.

Reviewer 2



The paper is focused on a very important problem of DNNs quantization. The authors propose a method of quantization of gradients, activations, and weights to 8-bit without a drop in test accuracy. The authors noticed that layer gradients do not follow a Gaussian distribution and connected this observation with the poor performance of low precision training. Based on this observation the authors suggested replacing several 8-bit matrix multiplications with 16-bit operations during the backward pass. It is necessary to note that 16-bit operations are applied only to such multiplications that do not involve a performance bottleneck. Another important component that leads to a good performance is Range Batch-Normalization (Range BN) operator. In other words, the authors introduced a more robust version of BN layer. Overall, it is a very interesting and well-written paper and the result is pretty strong. However, more experimental results on low-precision training without a drop in accuracy are required since it is the main contribution of the paper. The authors showed that their method has the same accuracy as a full-precision model only for ResNet-18 on ImageNet. Supplementary contains more experiments on more aggressive and, as a result, lossy quantization. Theoretical results in section 5 also contain several shortcomings: 1. In subsection 5.4 the authors’ reasoning is based on the fact that vectors W and eps are independent. However, components of eps are drawn from the uniform distribution with parameters dependent on max_i|W|_i. 2. In (13) inequality should be replaced with approximate inequality, since the authors consider approximation. In (12) the equality should be replaced with approximate equality due to the same reason. 3. To prove (9) the authors use Jensen's inequality. The classic Jensen's inequality has the form f(\mathbb{E} X) \leq \mathbb{E}f(x), where f is convex. In this work, the authors apply this inequality to get the inequality (9) of the form \mathbb{E} Y f(\mathbb{E}X) \leq \mathbb{E}(f(X) Y), where X and Y are not independent variables and f is convex. In other words, could you please elaborate how exactly you apply Jensen's inequality in (9), because under expectation in (9) there are two dependent variables (X = \|w\|_2 and Y = \|w\|_1) and f = 1 / x takes as the argument only one of these variables ( f(X) = 1 / \| w\|_2)? Update: I would like to thank the authors for their feedback. Since the authors provided new experimental results, I will change my score. However, I think that the theoretical part should be improved. ‘We note that unlike [1] which established that this angle converges to 37 degrees only at the limit when the dimension of the vector goes to infinity, our proof shows that this is a fundamental property valid also for the more practical case of finite dimensions.’ In the feedback, the authors state that the Jensen’s inequality is a good approximation. However, it is a good approximation when dimensionality is large enough (in other words it goes to infinity). Therefore this statement significantly resembles previous results. Moreover, since the authors use an approximation, they should not use the equality sign because it confuses readers.

Reviewer 3



At the hightest level, the manuscript contains three contributions: 1. Range BN. This is a clever way to make BN more low precision friendly. The authors motivate it thoroughly and back it up with convincing experiments. A good contribution 2. A largely heuristically derived training scheme that uses 8-bit weights and activations, a low (8-bit) and high (16-bit) copy of the deltas and some other tricks such as stochastic rounding. They show ImageNet ResNet18 to 65% accuracy matching full precision, which is a solid experimental validation of the method. 3. A lengthy section that tries to predict if a certain quantization scheme will be successful based on the cosine similarity between the original and the quantized weights. Overall, it appears to be a well-written, though out paper. Using INT8 for training is very novel and useful. For me the weak part of the paper is clearly Section 5. The hypothesis that angles are predictive of performance seems quite disconnected from the rest of the paper. I don't see evidence in the rest of the paper that this analysis provides anything predictive that could guide an experimenter. At the same time, it's not a theoretical result that is impactful on it's own. It seems like there might be possible ways the authors could have tried to make more of this analysis, for example injecting angle noise into a network and measuring how it affects performance. But there is neither intuition nor data what a certain angle means. What does it tell us if the angle for binary is 37°, which is almost right in the middle between no noise and orthogonal 90°? It's obvious that some error (in angle or by any other measure) will degrade performance, but to make this useful, there needs to be a functional relationship between the two. The relationship in Fig 2a) is tenuous at best, with a gradual fall-off in cosine and a sharp knee in accuracy. The derivation of the 2^M >> sqrt(ln(N)) condition is nicely done and promising, but the authors do not show that it's useful, which they could do e.g. by predicting the required number of bits for various operations and then showing convergence at this number of bits. Instead, they just observer that using 8 bits leaves a huge amount of headroom, rather than probing how tight this error bound is. My second, minor concern is that clarity should be improved. It's not exactly clear to me what data formats are used for which computations. It's laudable that the authors provide code to check, but this should be clear from the paper. Is the 8-bit format an integer format or fixed point, and how are exponents / scaling factors determined? What's the 16-bit format used for bifurcated gradients, is it fp16 or int16? Am I understanding correctly that weights and activations are in 8-bit and the gradients have 8- and 16-bit copies, or is there also a higher precision copy of the weights (common in many other low precision schemes)? What exactly do you mean by fp32 updates, does this imply there is a 32 bit copy of the weights, and if yes, where does stochastic rounding come in? This is partially addressed in the last paragraph, but should come earlier, and more clearly. Consider explaining it in form of a table, or better yet a diagram showing the various tensors that go into computing and updating a layer, and what type they are in. Misc comments: - The section around lines 79 is incorrectly claiming that BatchNorm has not successfully been applied in 16 bit. Köster et al. 2017 (Flexpoint: An Adaptive Numerical Format for Efficient Training of Deep Neural Networks) have shown that a ResNet-110 with BN can be trained completely in 16 bit, with no high precision variance parameters. - The statement in line 80 is not clear, is 42% supposed to be the overall SotA or for alexnet? It doesn't seem to be the right number for either - Can you elaborate on the serial dependence in line 140? Why does this matter? Does this assume specialized hardware where serial dependencies are more costly than raw FLOPS? - Figure 1: Caption should mention that this figure is for the ternary case - Figure 2a: Needs axis labels. bits on the horizontal and angle / accuracy on the vertical? - Fig2 caption: type, "with respect" should be "with respect to". Middle histogram in log scale -> all 3 are in log scale. - line 238, 10.5% for Cifar10 ResNet50. This model should train to about 93% accuracy according to He 2016. Please provide details on exact model used and why it's so much worse. - 245 typo "precising" In conclusion, the paper is solidly borderline, by which I mean that a rebuttal that addresses these concerns would make me reconsider the rating. ----------- Response to author rebuttal: In response to the rebuttal and discussion with other reviewers I'd like to update my assessment, the paper should be accepted. The new experimental results make the paper much stronger, and the authors were able to clear up a lot of misunderstandings, which I hope will make it through to the final version of the paper (e.g. a figure showing the elements of a layer, indicating the data type for each operation)